# Epidemiological trends of Lassa fever in Nigeria, 2018–2021

**Mahmood M. Dalhat**[1,2]*, **Adebola Olayinka**[1,3], **Martin M. Meremikwu**[4,5], **Chioma Dan-Nwafor**[1,3], **Akanimo Iniobong**[1], **Lorretta F. Ntoimo**[6], **Ikenna Onoh**[1], **Sandra Mba**[1], **Cornelius Ohonsi**[1,3], **Chinedu Arinze**[1], **Ekpereonne B. Esu**[5,7], **Obinna Nwafor**[1], **Ipadeola Oladipupo**[8], **Michael Onoja**[1], **Elsie Ilori**[1], **Friday Okonofua**[9], **Chinwe L. Ochu**[1,3], **Ehimario U. Igumbor**[3,10], **Ifedayo Adetifa**[1]

**1** Nigeria Centre for Disease Control and Prevention, Abuja, Nigeria, **2** Infectious Diseases Control Centre, Kaduna State, Kaduna, Nigeria, **3** Nigeria COVID-19 Research Coalition, Abuja, Nigeria, **4** Department of Paediatrics, University of Calabar Teaching Hospital, Calabar, Nigeria, **5** Cochrane Nigeria, Institute of Tropical Diseases Research and Prevention, University of Calabar Teaching Hospital, Calabar, Nigeria, **6** Department of Demography and Social Statistics, Faculty of Social Sciences, Federal University Oye-Ekiti, Oye, Nigeria, **7** Department of Public Health, College of Medical Sciences, University of Calabar, Calabar, Nigeria, **8** Department of Statistics, University of Ilorin, Ilorin, Nigeria, **9** Centre of Excellence in Reproductive Health Innovation, University of Benin, Benin City, Nigeria, **10** Centre for Infectious Disease Research, Nigerian Institute of Medical Research, Lagos, Nigeria

* mmdalhat@gmail.com

**Data Availability Statement:** The deidentified, cleaned, coded dataset could be obtained from the Zenodo repository. Access to the dataset can be obtained through the URL: https://zenodo.org/

## Abstract

### Background

Lassa fever is a viral haemorrhagic fever endemic in Nigeria. Improved surveillance and testing capacity have revealed in an increased number of reported cases and apparent geographic spread of Lassa fever in Nigeria. We described the recent four-year trend of Lassa fever in Nigeria to improve understanding of its epidemiology and inform the design of appropriate interventions.

### Methods

We analysed the national surveillance data on Lassa fever maintained by the Nigeria Centre for Diseases Control (NCDC) and described trends, sociodemographic, geographic distribution, and clinical outcomes. We compared cases, positivity, and clinical outcomes in the period January 2018 to December 2021.

### Results

We found Lassa fever to be reported throughout the year with more than half the cases reported within the first quarter of the year, a recent increase in numbers and geographic spread of the virus, and male and adult (>18 years) preponderance. Case fatality rates were worse in males, the under-five and elderly, during off-peak periods, and among low reporting states.

### Conclusion

Lassa fever is endemic in Nigeria with a recent increase in numbers and geographical distribution. Sustaining improved surveillance, enhanced laboratory diagnosis and improved

record/7309567#.Y23a5XbMI2x or DOI: 10.5281/zenodo.7309567.

**Funding:** The research was funded by the Department of Health and Social Care using UK Aid funding and is managed by the National Institute for Health and Care Research. The funders had no influence on the study design, data collection, analysis, decision to publish, or preparation of the manuscript.

**Competing interests:** The authors declared no competing interests.

case management capacity during off-peak periods should remain a priority. Attention should be paid to the very young and elderly during outbreaks. Further research efforts should identify and address specific factors that determine poor clinical outcomes.

## Introduction

Lassa fever is a neglected tropical disease that is endemic in West Africa. It has important global health implications given that it is the most exported of all the viral haemorrhagic fevers (VHFs), including Ebola [1–3]. The causative agent of this zoonotic acute VHF is a single-stranded ribonucleic acid (RNA) virus belonging to the Arenaviridae family. The major reservoir of Lassa virus is the multimammate rat of the genus Mastomys; also been discovered in other rodents [4–7]. It accounts for an estimated two million infections, 300,000–500,000 clinical infections, and 10,000 deaths yearly among inhabitants of the West African sub-region, where it remains endemic [8–12]. It confers a serious burden in endemic areas where it accounts for 6.0% of fevers, 0.7% of hospital admissions, with 40% case fatality, and almost a quarter of maternal mortality during peak periods [13–18]. Infection occurs following exposure to food or household items contaminated with the excreta or urine from infected rodents, or via person-to-person transmission through unprotected contact with body fluids, a common cause of healthcare workers (HCWs) infection [14, 19, 20].

Identified local practices that fuels Lassa fever infection and further transmission includes exposure to food or surfaces contaminated with droppings or urine of infected rodents–often the result of open drying of grains, processing of infected rats for consumption, and direct human-human transmission through close contact in community settings with prevailing poor infection, prevention and control (IPC) measures [21, 22].

Outbreaks occur frequently with a peak during the dry season months of November to April [23].

Before 2016, there was limited capacity for laboratory diagnosis of Lassa fever in Nigeria. The diagnosis was predominantly partner supported in the Institute of Lassa Fever Research and Control (ILFR&C), Irrua Specialist Teaching Hospital (ISTH), Nigeria, and the Lagos University Teaching Hospital (LUTH) laboratory, among the few specialized Lassa fever centers in the West African region. Evaluation of cases referred to the hospital revealed year-round transmission with peak levels between January and March [23]. Previous attempts at describing Lassa fever were limited by a lack of robust, nationwide, case-based data [23].

From 2016 until date, the Nigeria Centre for Disease Control (NCDC) has worked with various institutions to improve the diagnostic capacity for Lassa fever. At present, a national network of seven laboratories testing for the disease is being coordinated by the NCDC National Reference Laboratory (NRL). The seven laboratories are located at the Federal Teaching Hospital Abakaliki (FETHA); Irrua Specialist Teaching Hospital (ISTH); Lagos University Teaching Hospital (LUTH); Federal Medical Centre Owo; National Reference Laboratory (NRL) Gaduwa; Bayero University Teaching Hospital Kano; and Abubakar Tafawa Balewa University Teaching Hospital Bauchi.

By 2018, the NCDC supported the states to commence comprehensive documentation of Lassa fever through an electronic case-based surveillance system—Surveillance Outbreak Response Management and Analysis System (SORMAS). These efforts improved Lassa fever surveillance with increase in reported cases culminating in the largest ever recorded outbreak in the world, with increased incidence and geographic spread affecting 23 states (out of the

Nigerian 36 States and the Federal Capital Territory) [14]. Several plausible explanations were provided for the observed Lassa fever endemicity and the recent increase in the number of cases [24, 25]. In addition, the surveillance, testing, and reporting have significantly improved as highlighted above [13, 14]. Furthermore, possible redistribution of the reservoir and identification of additional hosts, as well as changing human behaviour and interaction with a consequent increase in contact with rodents are additional contributory factors [14, 26]. Prior to commencement of the case-based data provided by SORMAS, there was no national level elucidation of the trend, geographic distribution, differences in mortality between age groups and regions, as well as exploration of the possible reasons for the differences in mortality. These are important information required to improve surveillance and response for Lassa fever. To enable further understanding of the epidemiology and trend of Lassa fever over the years, we analysed four-year Lassa fever surveillance data to inform the design of effective prevention and control strategies.

## Materials and methods

### Study setting

Nigeria administratively has 36 states and a Federal Capital Territory (FCT). The States are zoned to six geopolitical areas: South-South, South-West, South-East, North-East, North-West, and North-Central. Each State is divided into lower administrative levels, the Local Government Areas (LGAs) of which there are 774 across the 36 states and the FCT. The country typically has two seasons, the rainy season which starts in March and ends in November, and the dry season starts in December and ends in April. The provision of public health services are concurrent responsibilities of the local, state, and federal governments. Health care service delivery involves primary, secondary, and tertiary public health facilities, as well as private health facilities.

The State Epidemiologists, State Disease Surveillance and Notification Officers (DSNO), and LGA DSNO coordinate surveillance activities at the state and LGA levels respectively. Through these officers, the NCDC collects data in line with the Disease Surveillance and Notification system guided by the Integrated Disease Surveillance and Response Strategy in Nigeria [27].

### Study design

We conducted a retrospective analysis of Lassa fever surveillance data from all the states in Nigeria, including the FCT between January 2018 to December 2021. We used retrospective national routine Lassa fever surveillance data. It comprises data from laboratory, case management (clinical) and case investigation (epidemiological) forms. The forms capture demographic information (age, sex, residential address, occupation), date of admission, hospital name, clinical details (date of symptom onset, symptoms) and exposures (contact with known or suspected Lassa fever case, being part of a contact tracing list, history of travel, direct contact with rodents or rodent faeces and urine, participation in burial activity) among other relevant epidemiological variables. All suspected Lassa fever cases reported to the DSNO for each LGA and the State Epidemiologist for each state were investigated using the Lassa fever case investigation form (CIF). Blood samples were collected and tested for all suspected cases. All suspected, probable, and confirmed cases were line-listed, and the information in the CIFs was uploaded in real-time on SORMAS, the primary digital platform for implementing the Integrated Diseases Surveillance and Response (IDSR) in Nigeria, used to generate the weekly Lassa fever situation reports (SITREPs). Currently, all states across the country use this platform to collect surveillance data, which is processed on a central server at the NCDC

headquarters in Abuja, Nigeria. This provides the most robust and representative Lassa surveillance data ever collected in the country with significantly improved key attributes like accuracy, timeliness, and completeness as well as additional sociodemographic and case outcome data. This provides more robust country-level, case-based data for more detailed epidemiological analyses evidenced by the increased geographical spread and absolute numbers reported [14, 24, 25]. We minimized the possibility of misclassification affecting the validity of our findings limiting key findings, conclusions and recommendations to confirmed cases rather than suspected cases.

For evaluation of the trend, we used the data from the NCDC Lassa fever Situation Reports (SITREP) complemented by laboratory testing data from the NCDC network of laboratories for the period January 2018 to December 2021. For further descriptive epidemiology, we used the 2018–2021 anonymized Lassa fever case-based surveillance data from SORMAS.

## Laboratory confirmation

Ethylenediaminetetraacetic acid (EDTA) bottles were used to collect blood samples from all suspected Lassa fever cases in the inpatient or outpatient departments of health facilities. Neonates were tested regardless of whether they exhibited symptoms if the mother was a confirmed case of Lassa fever. Samples were triple packaged and aseptically transported in a cold chain to any of the seven designated Lassa fever laboratories in Nigeria for the confirmatory test by reverse transcriptase-polymerase chain reaction (RT-PCR).

## Definition of cases and variables

**Case definition.** We defined Lassa fever cases based on the case definition provided in the IDSR guidelines as follows:

**Suspected case.** Any individual presenting with one or more of the following: malaise, fever, headache, sore throat, cough, nausea, vomiting, diarrhoea, myalgia, chest pain, hearing loss and either:

1. history of contact with excreta or urine of rodents,
   or

2. history of contact with a probable or confirmed Lassa fever case within a period of 21 days of onset of symptoms or any person with inexplicable bleeding.

**Confirmed case.** Any suspected case with laboratory confirmation (positive IgM antibody, PCR or virus isolation). RT-PCR was used in the diagnosis of all suspected cases.

**Probable case.** Any suspected case who died or absconded before collection of specimens for laboratory testing.

**Contact.** Any person who has been exposed to an infected person, or an infected person's secretions, excretions, or tissues within three weeks of the last contact with a confirmed or probable case of Lassa fever.

**Epidemiological week.** We defined the first epidemiological week of the year as the week that ends on the first Saturday of January, as long as it falls at least 4 days into the month. Subsequently each epidemiological week begins on a Sunday and ends on a Saturday.

**Healthcare workers.** We defined healthcare workers as all personnel working in health facilities regardless of the type of duties they perform.

**Case positivity rate.** The number of confirmed cases divided by the number of suspected cases tested multiplied by 100.

**Case fatality rate.** The number of deaths divided by the number of confirmed cases multiplied by 100.

**High-burden states.** For the purpose of this study, we defined high-burden states as states that consistently reported at least three-quarters of the cases for a given year.

## Data management and analyses

The data that support the findings of this study are available from NCDC, but restrictions apply to the availability of these data. Anonymized clinical and epidemiologic data of cases are available on request, conditional on the recipient agreeing to the NCDC data sharing and use guidelines.

All data cleaning and analyses were carried out in SPSS version 24. We calculated the proportion of missing data to assess completeness. We adopted a complete-case approach to analyze the socio-demographic variables. Socio-demographic and clinical characteristics of the study participants were described in terms of frequencies and percentages (%) for binary/categorical variables, and with mean and standard deviation (SD) for continuous variables, unless otherwise indicated.

We used dates of laboratory confirmation to plot the line graph (EpiCurve) showing the trend of cases over the four-year period (2018–2021). We considered an epidemiological week to begin on Sunday and end on Saturday. We furthermore calculated age-sex distribution, age-specific case fatality rates (CFRs) and test positivity for the different years studied.

## Ethical considerations

The protocol for this study was reviewed and approved by the Nigerian National Health Research Ethics Committee (Approval Number: NHREC/01/01/2007-27/04/2022) with exemption from informed consent as the data was collected, stored and analysed part of outbreak response activity by the NCDC from which permission to use was obtained. All data were kept confidential and stored in password-protected computers. Personal identifiers were not extracted, and the dataset was anonymised before sharing.

## Results

We documented 3,162 confirmed cases with complete information on clinical outcomes, and 550 deaths (CFR = 17.4%) (see Table 1). Based on a complete-case approach to the analysis of socio-demographic variable (sex) with missing data (1.4%), 1,394 (44.7%) were females. Civil servants and farmers appeared to be more affected with double digit percentages compared to other occupations. Case fatality rates were generally higher in males except for 2020 when females had marginally higher CFR, worst in 2018 compared to other years, and higher among the very young and very old age groups (Table 2). Furthermore, CFR had significant improvement over the years in high-burden states except for 2021.

Fig 1 shows the monthly distribution of cases between 2018 and 2021. Lassa fever was reported all year round with the highest number of cases from the last quarter of the preceding year through to the first quarter, with the highest outbreak occurring in 2020. For each year, the peaks were reached within epidemiological weeks 4–10. Subsequently, cases were recorded consistently at a lower level for most of the year. Notably, in 2021, the Lassa fever outbreak did not peak as much as it did in the two previous years (2019 and 2020).

Age and sex distribution revealed slight male preponderance that consistently decreased through the years and fair distribution among the age groups except for the 0–9 age group who have less than half of the cases seen in the other age groups. The highest proportion of confirmed cases in the years reviewed was contributed by students, civil servants, and traders;

**Table 1. Sociodemographic characteristics of confirmed cases, 2018–2021.**

| Variable | 2018 N (%) | 2019 N (%) | 2020 N (%) | 2021 N (%) |
|---|---|---|---|---|
| Sex (n = 3118) | | | | |
| Female | 151 (40.3) | 394 (44.3) | 610 (45.1) | 239 (47.7) |
| Male | 224 (12.99) | 496 (28.77) | 742 (43.04) | 262 (15.20) |
| Age group (n = 3162) | | | | |
| 0–4 | 6 (1.6) | 36 (4.0) | 51 (3.7) | 14 (2.8) |
| 5–9 | 3 (0.8) | 10 (1.1) | 11 (0.8) | 3 (0.6) |
| 10–19 | 53 (13.7) | 107 (11.8) | 148 (10.8) | 77 (15.4) |
| 20–29 | 79 (20.4) | 197 (21.8) | 317 (23.2) | 109 (21.8) |
| 30–39 | 89 (23.0) | 184 (20.3) | 254 (18.6) | 83 (16.6) |
| 40–49 | 48 (12.4) | 133 (14.7) | 211 (15.4) | 83 (16.6) |
| 50–59 | 47 (12.1) | 82 (9.1) | 131 (9.6) | 37 (7.4) |
| 60+ | 62 (16.0) | 156 (17.2) | 246 (18.0) | 95 (19.0) |
| Occupation (N = 3162) | | | | |
| Artisan | 5 (1.3) | 24 (2.7) | 50 (3.7) | 8 (1.6) |
| Child/Pupil | 24 (6.2) | 99 (10.9) | 105 (7.7) | 32 (6.4) |
| Civil/Public Servant | 57 (14.7) | 314 (34.7) | 294 (21.5) | 98 (19.6) |
| Farming/Livestock | 36 (9.3) | 65 (7.2) | 96 (7.0) | 27 (5.4) |
| Health Worker | 16 (4.1) | 19 (2.1) | 30 (2.2) | 17 (3.4) |
| Housewife | 8 (2.1) | 39 (4.3) | 56 (4.1) | 10 (2.0) |
| Religious Leader | 6 (1.6) | 6 (0.7) | 17 (1.2) | 1 (0.2) |
| Retiree | 3 (0.8) | 8 (0.9) | 15 (1.1) | 12 (2.4) |
| Student | 44 (11.4) | 136 (15.0) | 218 (15.9) | 90 (18.0) |
| Teacher/Lecturer | 7 (1.8) | 7 (0.8) | 76 (5.6) | 8 (1.6) |
| Trader | 42 (10.9) | 118 (13.0) | 197 (14.4) | 164 (32.7) |
| Unemployed | 131 (33.9) | 37 (4.1) | 171 (12.5) | 22 (4.4) |
| Other | 8 (2.1) | 33 (3.6) | 44 (3.2) | 12 (2.4) |
| State of Residence (N = 3162) | | | | |
| Edo | 236 (61.0) | 522 (57.7) | 583 (42.6) | 221 (44.1) |
| Ondo | 102 (26.4) | 216 (23.9) | 295 (21.5) | 143 (28.5) |
| Ebonyi | 8 (2.1) | 43 (4.8) | 81 (5.9) | 35 (7.0) |
| Bauchi | 13 (3.4) | 21 (2.3) | 88 (6.4) | 30 (6.0) |
| Plateau | 1 (0.3) | 3 (0.3) | 39 (2.8) | 17 (3.4) |
| Taraba | 6 (1.6) | 9 (1.0) | 44 (3.2) | 19 (3.8) |
| Other States | 28 (7.2) | 103 (11.4) | 322 (23.5) | 72 (14.4) |
| Date of Report in Quarters (N = 3162) | | | | |
| 1st Quarter | 209 (54.0) | 479 (52.9) | 806 (58.9) | 174 (34.7) |
| 2nd Quarter | 61 (15.8) | 208 (23.0) | 210 (15.3) | 35 (7.0) |
| 3rd Quarter | 19 (4.9) | 119 (13.1) | 233 (17.0) | 56 (11.2) |
| 4th Quarter | 98 (25.3) | 99 (10.9) | 120 (8.82) | 136 (47.1) |
| Clinical Outcome | | | | |
| Survived | 298 (77.0) | 769 (85.0) | 1137 (83.1) | 408 (81.4) |
| Dead | 89 (23.0) | 136 (15.0) | 232 (16.9) | 93 (18.6) |

the other sub-groups have lower numbers in comparison. Health workers as a subset contributed 4.1% for the year 2018, the highest for any of the four years under study. There does not appear to be consistency in the proportion of cases contributed by students whose proportion was seen to be high in 2019 and 2020 compared to the other years (Table 1).

**Table 2. Case fatality rates by key socio-demographic characteristics, 2018–2021.**

| | Deaths (CFR) | | | |
|---|---|---|---|---|
| **Variable** | **2018** | **2019** | **2020** | **2021** |
| Sex | | | | |
| Female | 27 (17.9) | 53 (13.5) | 106 (17.4) | 36 (15.1) |
| Male | 58 (25.9) | 82 (16.5) | 123 (16.6) | 57 (21.8) |
| Age group | | | | |
| 0–4 | 2 (33.3) | 6 (16.7) | 12 (23.5) | 4 (28.6) |
| 5–9 | 1 (33.3) | 1 (10.0) | 1 (9.1) | 0 (0.0) |
| 10–19 | 10 (18.9) | 11 (10.3) | 13 (8.8) | 13 (16.9) |
| 20–29 | 18 (22.8) | 14 (7.1) | 41 (12.9) | 14 (12.8) |
| 30–39 | 25 (28.1) | 23 (12.5) | 40 (15.8) | 18 (21.7) |
| 40–49 | 9 (18.8) | 26 (19.6) | 35 (16.6) | 13 (15.7) |
| 50–59 | 10 (21.3) | 17 (20.7) | 30 (22.9) | 10 (27.0) |
| 60+ | 14 (22.6) | 38 (24.4) | 60 (24.4) | 21 (22.1) |
| Occupation (N = 3162) | | | | |
| Artisan | 1 (20.0) | 5 (20.8) | 12 (24.0) | 5 (62.5) |
| Child/Pupil | 5 (20.8) | 9 (9.1) | 21 (20.0) | 8 (25.0) |
| Civil/Public Servant | 9 (15.8) | 51 (16.3) | 37 (12.6) | 18 (18.4) |
| Farming/Livestock | 10 (27.8) | 18 (27.7) | 34 (35.4) | 9 (33.3) |
| Health Worker | 2 (12.5) | 1 (5.3) | 5 (16.7) | 4 (23.5) |
| Housewife | 4 (50.0) | 11 (28.2) | 19 (33.9) | 2 (20.0) |
| Religious Leader | 1 (16.7) | 1 (16.7) | 3 (17.7) | 0 (0.0) |
| Retiree | 2 (66.7) | 3 (37.5) | 3 (20.0) | 5 (41.7) |
| Student | 6 (13.6) | 6 (4.4) | 17 (7.8) | 11 (12.2) |
| Teacher/Lecturer | 1 (14.3) | 1 (14.3) | 13 (17.1) | 3 (37.5) |
| Trader | 11 (26.2) | 19 (16.1) | 42 (21.3) | 23 (14.0) |
| Unemployed | 34 (26.0) | 6 (16.2) | 17 (9.9) | 5 (22.7) |
| Other | 3 (37.5) | 5 (15.2) | 9 (20.5) | 0 (0.0) |
| State of Residence (N = 3162) | | | | |
| Edo | 53 (22.5) | 78 (14.9) | 106 (18.2) | 39 (17.7) |
| Ondo | 21 (20.6) | 39 (18.1) | 45 (15.3) | 23 (16.1) |
| Ebonyi | 3 (37.5) | 5 (11.6) | 12 (14.8) | 8 (22.9) |
| Bauchi | 1 (7.7) | 5 (23.8) | 13 (14.8) | 6 (20.0) |
| Plateau | 1 (100) | 0 (0.0) | 11 (28.2) | 4 (23.5) |
| Taraba | 1 (16.7) | 0 (0.0) | 10 (22.7) | 5 (26.3) |
| Other States | 9 (42.9) | 9 (9.9) | 35 (14.6) | 8 (22.2) |
| Date of Report in Quarters (N = 3162) | | | | |
| 1st Quarter | 50 (23.9) | 87 (18.2) | 146 (18.1) | 28 (16.1) |
| 2nd Quarter | 11 (18.0) | 26 (12.5) | 24 (11.4) | 10 (28.6) |
| 3rd Quarter | 4 (21.1) | 13 (10.9) | 43 (18.5) | 13 (23.2) |
| 4th Quarter | 24 (24.5) | 10 (10.1) | 19 (15.8) | 42 (17.8) |

In terms of state-level distribution, except for the year 2020, at least 75.0% of cases were reported from 4 States (Edo, Ondo, Ebonyi, and Bauchi). In terms of seasonality, almost two-thirds of the cases were reported within two quarters (the 4th quarter of the preceding year and the 1st quarter of the current year). These 2 quarters accounted for 79.0%, 64.0%, 68.0% and 82.0% of confirmed cases reported in 2018, 2019, 2020 and 2021 respectively (Table 1).

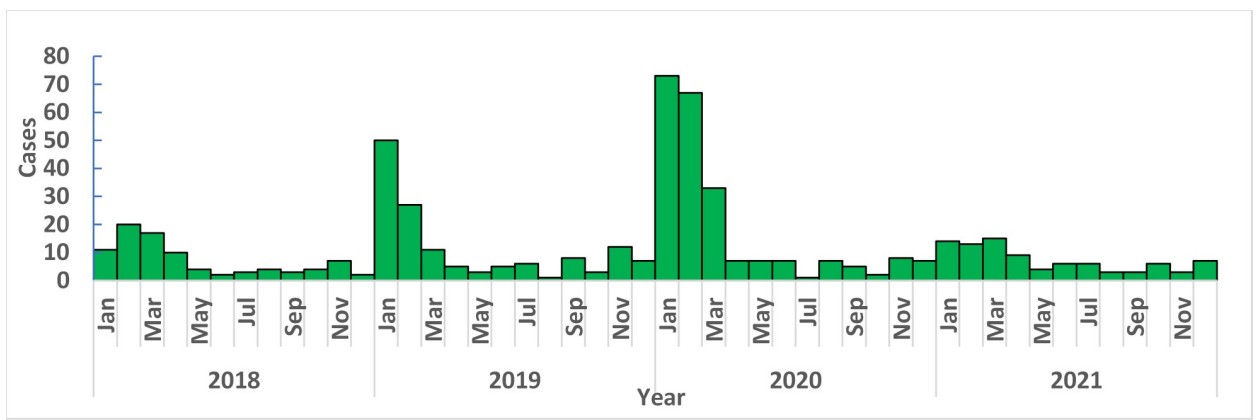

**Fig 1. Epicurve of monthly distribution of Lassa fever cases in Nigeria, 2018–2021.**

The test positivity rate was uniformly high in 2018 albeit with alternating peaks and troughs. Except for 2021, the test positivity rate appeared to have risen to the highest level at the beginning of the peak periods (epi week 4) and then descended to a lower level and remained grossly consistent for most of the year. There is decrease in case positivity observed with increase in the number of cases (Fig 2).

Fig 3 shows the weekly trend in number of confirmed cases, deaths, and CFRs. Case fatality rates were observed to be inversely proportional to the number of cases with the highest peaks seen when cases were low. Thus, CFRs were lowest in the first quarter of the year when number of cases was highest (Fig 3).

Case density for Lassa fever across the States has consistently increased from 2018 (Fig 4A) when 394 cases were reported across 20 States and 93 LGAs, through 2019 (Fig 4B) when 917 cases were reported across 23 States and 86 LGAs, to 2020 (Fig 4C) when 1452 cases were reported across 29 States and 131 LGAs. The year 2021 showed numbers and geographical distribution of Lassa fever cases below 2018 levels with 537 cases reported across 16 States and 63 LGAs (Fig 4D).

## Discussion

We described the four-year trend of Lassa fever in Nigeria following improved surveillance and laboratory diagnosis culminating in the largest outbreaks of Lassa fever ever reported. Our

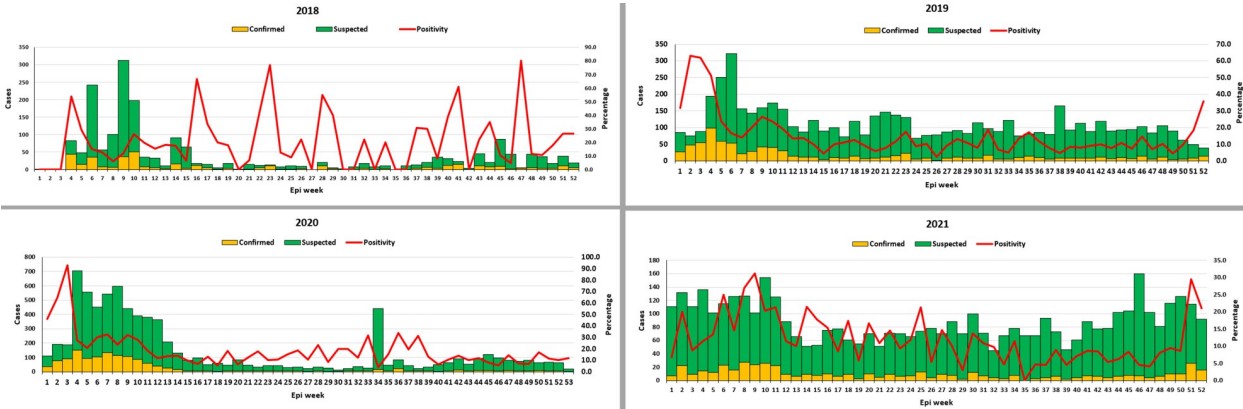

**Fig 2. Trend of weekly distribution of Lassa fever suspected and confirmed cases as well as the test positivity rates, Nigeria, 2018–2021.**

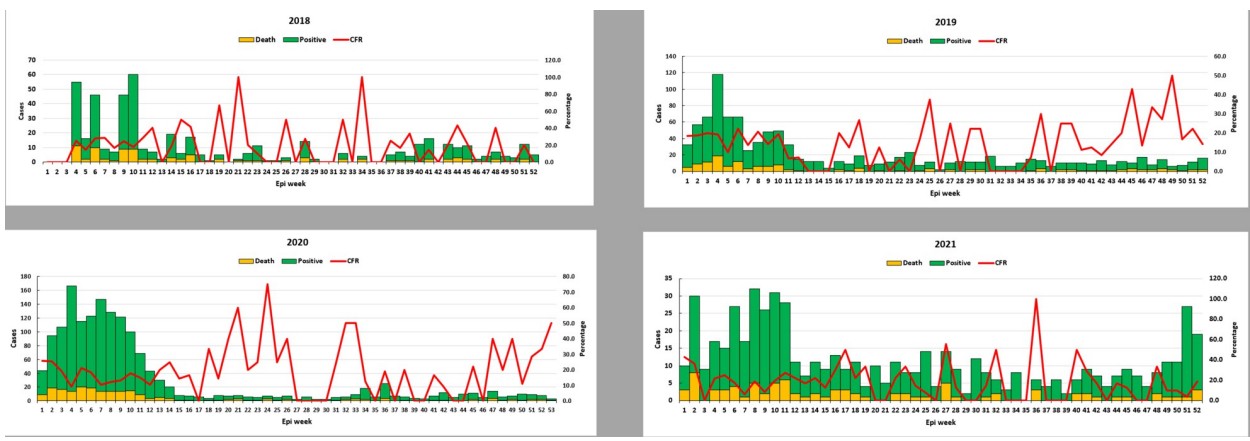

**Fig 3. Weekly distribution of confirmed cases, deaths, and case fatality rates, 2018–2022.**

data show a general increase in the case density and geographical distribution of Lassa fever across the States from the year 2018 through 2020. This has been noted by earlier works describing the possible changing epidemiology of Lassa fever in Nigeria [13, 23]. In 2021, the number of Lassa fever cases declined towards levels observed in 2018 albeit with more limited geographical distribution. The increase in reported numbers and apparent geographic distribution seen in 2018 through 2020 might be partly explained by the improved capacity of the surveillance system due to the introduction of SORMAS and an increase in the numbers and distribution of Lassa fever testing laboratories in the country. The increased availability of testing laboratories or access to testing provided by the established sample transport system to the NCDC network of laboratories might have given healthworkers more motivation to report and provide samples for testing and thus, the reason for more cases seen in hitherto non-traditional hotspots. The drop in the reported numbers for 2021 is likely attributable to either the

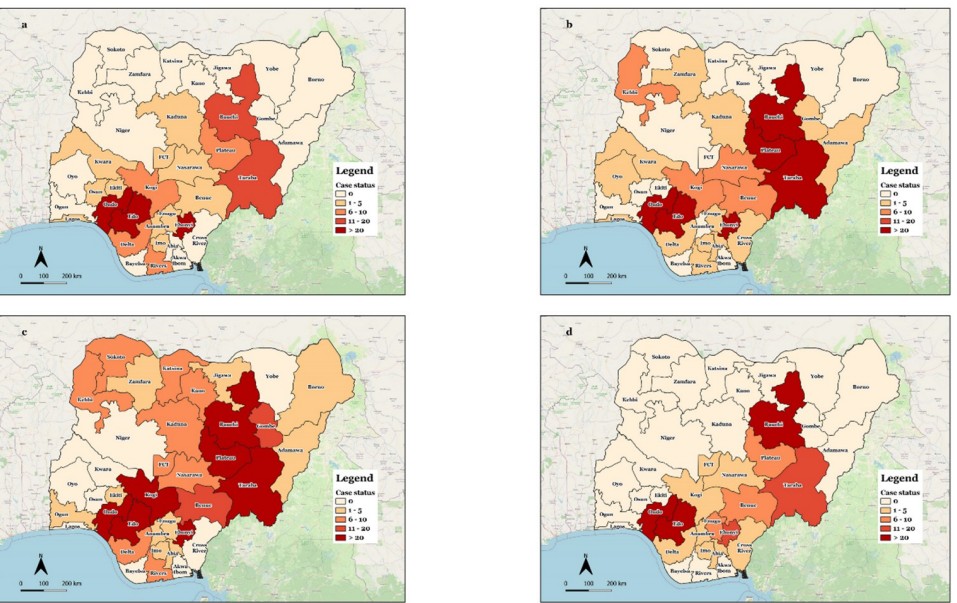

**Fig 4.** Distribution of Lassa fever cases by state, 2018–2021 (2018 {4a}; 2019 {4b}; 2020 {4c}; 2021 {4d}).

impact of fatigue on the surveillance system due to the massive outbreaks in 2018 and 2019 and/or diversion of surveillance and response resources to other equally important challenges like the COVID-19 pandemic; otherwise, it may reflect a real reduction in cases. An earlier review of the surveillance system in 2022 showed numbers that surpassed those reported in 2021 [28]. Importantly, the increased numbers and geographic distribution in 2019 and 2020 cannot be explained completely by improved surveillance and laboratory diagnosis alone given that despite the high numbers of suspected cases in 2021, there were significantly lower confirmed case counts compared to the unprecedented numbers seen in 2018–2020. Furthermore, the agent-related factor of changing virulence and pathogenicity of the virus is also an unlikely contributor given that previous effort to explore this has shown no significant change in the genealogy of the virus over the years [29]. There is nonetheless a need to sustain the momentum gained for surveillance that resulted from the 2018–2020 outbreaks.

The Lassa fever outbreaks appear to be more intense in the last quarter of the year through to the first quarter of the year after which cases continue to occur at a lower level for the rest of the year. This has been largely hypothesised to be the result of increased population of rodents due to more favourable condition for breeding towards the end of the rainy season and beginning of the dry season largely characterized by abundance of feeds on the farms [30–33]. The year-round reporting of cases further confirms the endemicity of Lassa fever in Nigeria and the need to focus on hitherto unrecognized endemic hotspots for optimised clinical evaluation, testing, confirmation, and treatment as well as the development and deployment of rapid diagnostic tests (RDTs) for routine evaluation of febrile illnesses. Furthermore, active deployment of preparedness and response resources towards the fourth quarter of the year as well as sustaining vigilance over the remaining months could improve early detection, confirmation, and case management thereby reducing morbidity and mortality.

The observed predominance of civil servants and farmers might be related to better health-seeking behaviour of the former and increased exposure to humans and food in the latter group. Unhygienic exposure of farm produce to rodents during processing in rural communities has been identified as an important factor in the rodents-human transmission of Lassa [21, 22].

The variation in case positivity rates in relation to the outbreaks implies more proficiency at the beginning of outbreaks when clinicians were more alert, especially in areas that commonly diagnose and manage cases of the disease. The subsequent decline in case positivity might be the low threshold to screen cases of fever that results from increased active or passive sensitization of healthcare workers. This calls for further advocacy, training, and sensitization of healthcare workers, especially in-between outbreaks.

The observed predominance of males among confirmed cases might reflect more exposure of the males to risk factors of Lassa fever or difference in access to care and thus opportunity to be evaluated and tested. However, the latter may not be plausible given the observed higher CFRs seen in men. As in previous studies, the extremes of ages showed higher CFR implying that the known vulnerability relating to these age groups is true even for Lassa fever [34, 35].

The observed inverse relationship between the number of cases and case-fatality rates may be the result of the deployment of more resources during large outbreaks (human and financial resources, political will, activation of EOC/IMS) or the disparity of health workers' proficiency between high-density states and areas with low density. As expected, those providing clinical care in endemic areas will be more proficient and will have a lower threshold for suspecting and requesting tests thereby reducing delayed diagnosis and commencement of treatment. The fact that there is a dedicated centre, the Institute of Lassa Fever Research and Control (ILFR&C), Irrua Specialist Teaching Hospital (ISTH) within or in close proximity to the high-burden States might further explain the differential in mortality. There is thus, a need to

further investigate the reasons for lower detection rates and high case fatality in the so-called 'non-endemic' states reporting lower cases with aim of identifying and implementing targeted interventions.

Our study is limited by the fact that the observations made are based on the numbers reported without factoring in specific surveillance and response efforts that might explain variation in numbers of cases, positivity, and clinical outcomes. Also, our data was based on a facility-based rather than community-based surveillance system meaning that cases that did not seek hospital care, those that died at home, and asymptomatic cases were not captured. These factors will possibly lead to the observed high case fatality rate given that mild-moderate cases were not suspected and confirmed and consequently not part of the surveillance data used for this study. Finally, long-term efforts to control Lassa fever should include economic evaluations as well as infectious diseases modelling which are beyond the scope of this work but have been explored by earlier works [9, 36–40]. This will enable proactive rather than reactive responses to the recurrent outbreaks of Lassa as well as other viral haemorrhagic fevers.

We conclude that Lassa fever is endemic in Nigeria with an apparent increase in geographical spread owing, at least in part, to improved surveillance and reporting as well as laboratory diagnosis. We recommend efforts to evaluate and identify areas for further strengthening of the Lassa fever surveillance system to sustain the gains from the improvement in the surveillance system. We also recommend the optimization of human and financial resources for a response especially in between outbreaks. Furthermore, better testing capacity along with health worker training in States reporting low cases could lead to timely detection, prompt testing, early diagnosis, and more effective treatment and improved outcomes from Lassa fever cases. Additional studies are needed to identify determinants of prompt identification, testing, confirmation, and treatment of cases and consequently better clinical outcomes of Lassa fever cases. Prevention and prompt diagnosis and treatment among identified vulnerable groups like children under-five years and the elderly should be a priority during outbreaks to reduce the higher CFR in these groups.

## Acknowledgments

We appreciate the National Lassa Fever Technical Working Group, the epidemiology teams of Nigerian State Ministries of Health, Lassa fever treatment centres, Lassa fever testing laboratory network and all partners supporting the fight against Lassa fever.

## Author Contributions

**Conceptualization:** Mahmood M. Dalhat, Adebola Olayinka, Martin M. Meremikwu, Chioma Dan-Nwafor, Lorretta F. Ntoimo, Chinwe L. Ochu, Ehimario U. Igumbor.

**Data curation:** Lorretta F. Ntoimo, Sandra Mba, Chinedu Arinze, Obinna Nwafor.

**Formal analysis:** Akanimo Iniobong, Lorretta F. Ntoimo.

**Funding acquisition:** Ifedayo Adetifa.

**Investigation:** Chioma Dan-Nwafor, Sandra Mba, Ipadeola Oladipupo, Michael Onoja.

**Methodology:** Mahmood M. Dalhat, Chioma Dan-Nwafor.

**Project administration:** Chioma Dan-Nwafor, Cornelius Ohonsi, Chinwe L. Ochu.

**Software:** Mahmood M. Dalhat, Akanimo Iniobong.

**Supervision:** Adebola Olayinka, Elsie Ilori, Friday Okonofua, Chinwe L. Ochu, Ehimario U. Igumbor, Ifedayo Adetifa.

**Validation:** Mahmood M. Dalhat, Ikenna Onoh, Ekpereonne B. Esu, Elsie Ilori, Friday Okonofua.

**Visualization:** Akanimo Iniobong.

**Writing – original draft:** Mahmood M. Dalhat.

**Writing – review & editing:** Mahmood M. Dalhat, Adebola Olayinka, Martin M. Meremikwu, Chioma Dan-Nwafor, Akanimo Iniobong, Lorretta F. Ntoimo, Ikenna Onoh, Sandra Mba, Cornelius Ohonsi, Chinedu Arinze, Ekpereonne B. Esu, Obinna Nwafor, Ipadeola Oladi-pupo, Michael Onoja, Elsie Ilori, Friday Okonofua, Chinwe L. Ochu, Ehimario U. Igumbor, Ifedayo Adetifa.

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
