## [Decision Letter · Decision Letter 0]

2 Sep 2022

PONE-D-22-22331Epidemiological Trends of Lassa Fever in Nigeria, 2018 - 2021PLOS ONE

Dear Dr. Dalhat,

Thank you for submitting your manuscript to PLOS ONE. After careful consideration, we feel that it has merit but does not fully meet PLOS ONE’s publication criteria as it currently stands. Therefore, we invite you to submit a revised version of the manuscript that addresses the points raised during the review process.

We are glad that the reviewers found their work adequate and that it could be published after corrections.

We ask that you diligently consider the reviewers' suggestions and respond carefully in the rebuttal letter, indicating the changes made to avoid a new round of review.

We look forward to receiving your revised manuscript.

Kind regards,

Kovy Arteaga-Livias

Academic Editor

PLOS ONE

Journal Requirements:

2. In the ethics statement in the manuscript and in the online submission form, please provide additional information about the patient records/samples used in your retrospective study. Specifically, please ensure that you have discussed whether all data/samples were fully anonymized before you accessed them and/or whether the IRB or ethics committee waived the requirement for informed consent. If patients provided informed written consent to have data/samples from their medical records used in research, please include this information.

3. Please ensure that you have specified (1) whether consent was informed and (2) what type you obtained (for instance, written or verbal, and if verbal, how it was documented and witnessed). If your study included minors, state whether you obtained consent from parents or guardians. If the need for consent was waived by the ethics committee, please include this information.

"This research is funded by the Department of Health and Social Care using UK Aid funding and is managed by the National Institute for Health and Care Research. The views expressed in this publication are those of the author(s) and not necessarily those of the Department of Health and Social Care."

"This research is funded by the Department of Health and Social Care using UK Aid funding and is managed by the National Institute for Health and Care Research. The funders had no influence on the study design, data collection, analysis, decision to publish, or preparation of the manuscript."

6. Thank you for stating the following in your Competing Interests section:  

"The authors declared no competing interests."

7. In your Data Availability statement, you have not specified where the minimal data set underlying the results described in your manuscript can be found. PLOS defines a study's minimal data set as the underlying data used to reach the conclusions drawn in the manuscript and any additional data required to replicate the reported study findings in their entirety. All PLOS journals require that the minimal data set be made fully available. For more information about our data policy, please see http://journals.plos.org/plosone/s/data-availability.

Reviewers' comments:

Reviewer's Responses to Questions

**Comments to the Author**

1. Is the manuscript technically sound, and do the data support the conclusions?

Reviewer #1: Yes

Reviewer #2: Yes

Reviewer #3: Yes

Reviewer #4: Yes

2. Has the statistical analysis been performed appropriately and rigorously? 

Reviewer #1: Yes

Reviewer #2: Yes

Reviewer #3: Yes

Reviewer #4: No

3. Have the authors made all data underlying the findings in their manuscript fully available?

Reviewer #1: No

Reviewer #2: Yes

Reviewer #3: Yes

Reviewer #4: Yes

4. Is the manuscript presented in an intelligible fashion and written in standard English?

Reviewer #1: Yes

Reviewer #2: Yes

Reviewer #3: Yes

Reviewer #4: Yes

5. Review Comments to the Author

Reviewer #1: After carefully reading the manuscript, I can say that the manuscript is technically sound, and the data support the conclusions. The data was not made available, however the authors mentioned this can be accessed on the NCDC website.

Reviewer #2: The paper brought out further evidence of the recent changes to the epidemiology of Lassa in the country, particularly its increased all year round incidence, and a new perspective on the differences in risk groups. Furthermore, being a review of country level secondary data, the paper stimulates further research interests based on some interesting hypothesis.

Reviewer #3: In general, the paper provides useful information pertaining to the current case distributions for Nigeria with regard to age, geographic distribution, and profession.

General comments:

Lines 266-267: Do environmental factors influence this increase in transmission during Q4 and Q1? Is there more human-human transmission during this time, or is there more direct rodent contact? Please provide rationale for this surge.

Additionally, please provide a description and a possible rationale for the uneven distribution of cases among professions. For example, are there more civil servants, students, and traders in general in the population, or is there actually a higher rate of exposure in these groups specifically?

Reviewer #4: The authors have done well in retrospectively analyzing reported Lassa fever cases in Nigeria, their work will be useful in guiding policy makers towards making decisions on preparedness plans. I have found their language to be clear and unambiguous. There are however some concerns that they need to address as outlined below:

Line 73, who commenced this comprehensive documentation in SORMAS? States, Federal, hospitals?

Line 107, why the use of different data sources? What is the role of SORMAS as stated earlier in line 73?

Statement in lines 119-120 doesn’t align with what was earlier states in lines 103-7

Line 130, was there any reason why the case definitions on the IDSR were not used?

Line 154, is this a standardized definition or was it adapted from the preparedness guideline?

In Table 2, is the N still valid for the occupation, state of residence and date of quarter in report on said table?

Line 199, the paragraph talks about monthly distribution, it will be good to reflect when the highest outbreak occurred in line 199 with respect to month and not year

In line 200, the use of epi week as against the earlier monthly distribution may be misleading, except a clear description of how epi weeks first relates to months is given

In line 207 while talking of low numbers contributed by pupils/children, Is this low number in comparison to any subset? Better to compare age differently and describe findings from there and do same separately for occupation to avoid mis-interpretation

Line 219, did you merge the age groups?

Line 224, reconfirm the comment on slight increase in relation to figure 2

Line 232 for CFRs, charts were in epiweek and results should follow same pattern , except if the authors would firstly define the epi weeks as they align with different quarter of the year

Lines 238-9, the number of affected states and fig 4a and 4c seem not tally

Line 251, did it return to or it was lower than the geographical distribution of 2018?

Lines 295-6, does the IDSR system not provide for probable cases and are they not reported in the SORMAS? You earlier gave a definition of probable cases, were there no probable cases reported within the review period?

6. PLOS authors have the option to publish the peer review history of their article (what does this mean?). If published, this will include your full peer review and any attached files.

Reviewer #1: **Yes: **Mayowa M Ojo

Reviewer #2: **Yes: **Mohammed Abdulkarim Abdullahi

Reviewer #3: **Yes: **Slobodan Paessler

Reviewer #4: No

---

## [Author Response · Author response to Decision Letter 0]

17 Nov 2022

We have made point by point response to the comments and attached as instructed in the document named: 'Response to Reviewers'

---

## [Decision Letter · Decision Letter 1]

9 Dec 2022

Epidemiological Trends of Lassa Fever in Nigeria, 2018 - 2021

PONE-D-22-22331R1

Dear Dr. Dalhat,

We’re pleased to inform you that your manuscript has been judged scientifically suitable for publication and will be formally accepted for publication once it meets all outstanding technical requirements.

Kind regards,

Kovy Arteaga-Livias

Academic Editor

PLOS ONE

Additional Editor Comments (optional):

Reviewers' comments:

Reviewer's Responses to Questions

**Comments to the Author**

1. If the authors have adequately addressed your comments raised in a previous round of review and you feel that this manuscript is now acceptable for publication, you may indicate that here to bypass the “Comments to the Author” section, enter your conflict of interest statement in the “Confidential to Editor” section, and submit your "Accept" recommendation.

Reviewer #1: All comments have been addressed

Reviewer #2: All comments have been addressed

Reviewer #4: All comments have been addressed

2. Is the manuscript technically sound, and do the data support the conclusions?

Reviewer #1: (No Response)

Reviewer #2: Yes

Reviewer #4: Yes

3. Has the statistical analysis been performed appropriately and rigorously? 

Reviewer #1: (No Response)

Reviewer #2: Yes

Reviewer #4: Yes

4. Have the authors made all data underlying the findings in their manuscript fully available?

Reviewer #1: (No Response)

Reviewer #2: Yes

Reviewer #4: Yes

5. Is the manuscript presented in an intelligible fashion and written in standard English?

Reviewer #1: (No Response)

Reviewer #2: Yes

Reviewer #4: Yes

6. Review Comments to the Author

Reviewer #1: (No Response)

Reviewer #2: The author did the due diligence in addressing all the issues raised in the previous submission. The revised submission is much improved and meets the required standard. I believe the paper will be valuable for further research on Lassa fever.

Reviewer #4: (No Response)

7. PLOS authors have the option to publish the peer review history of their article (what does this mean?). If published, this will include your full peer review and any attached files.

Reviewer #1: No

Reviewer #2: **Yes: **Mohammed A. Abdulkarim

Reviewer #4: No

---

## [Editor Report · Acceptance letter]

21 Dec 2022

PONE-D-22-22331R1 

Epidemiological trends of Lassa fever in Nigeria, 2018 – 2021 

Dear Dr. Dalhat:

I'm pleased to inform you that your manuscript has been deemed suitable for publication in PLOS ONE. Congratulations! Your manuscript is now with our production department. 

Kind regards, 

on behalf of

Dr. Kovy Arteaga-Livias 

Academic Editor

PLOS ONE